# Societal cost of day-care approach (DCA): A low-cost approach in urban and rural settings for management of childhood severe pneumonia in Bangladesh

**Marufa Sultana**[1,2]*, **Jennifer Watts**[1], **Nur H. Alam**[2], **Nausad Ali**[2], **Abu S. G. Faruque**[2], **Sabiha Nasrin**[2], **Mohammod Jobayer Chisti**[2], **George J. Fuchs**[3], **Niklaus Gyr**[4], **Tahmeed Ahmed**[2], **Julie Abimanyi-Ochom**[1], **Lisa Gold**[1]

1 Deakin Health Economics, Institute for Health Transformation, School of Health and Social Development, Faculty of Health, Deakin University, Geelong, Victoria, Australia, 2 Nutrition Research Division; icddr,b, Dhaka, Bangladesh, 3 College of Medicine and College of Public Health, University of Kentucky, United States of America, 4 University Hospital, Basel, Switzerland

* m.sultana@deakin.edu.au

## Abstract

### Objective

Childhood severe pneumonia is the leading cause of under-five deaths in Bangladesh. A new day-care management approach (DCA) was implemented in primary-level healthcare facilities in urban and rural areas of Bangladesh. Reliable cost estimates are important to determine the economic viability of the new management approach. The objective of this study were to estimate the mean societal cost per patient for a new Day-care approach (DCA) in managing childhood severe pneumonia, to assess cost variation in urban and rural healthcare settings, and to determine important cost predictors.

### Study design

This study was conducted alongside a cluster randomized trial conducted in Bangladesh Children diagnosed with severe pneumonia were enrolled between November 2015 and March 2019. Employing a bottom-up micro-costing approach from a societal perspective, detailed household and provider cost data were collected from sixteen intervention facilities (n = 16). Data collection involved structured questionnaires administered face-to-face with facility staff, interviews with parents/caregivers, and patient record reviews. Analysis measured mean cost and cost variation across socio-economic groups, facility location, clinical variables, and determined cost-sensitive parameters. A p-value of < 0.05 was considered as statistically significant level.

**Data availability statement:** The study collected background information on the children and their clinical records, which included identifiable information. Therefore, the data cannot be made publicly available due to the inclusion of sensitive details within the clinical records of each patient. Making such data publicly available could compromise patient privacy and would contravene the protocol approved by our research ethics boards. The ethical boards of icddr,b in Bangladesh and Deakin University in Australia, where the study was conducted, explicitly require that data containing sensitive or identifiable information be securely stored and accessed only by authorised personnel to safeguard patient confidentiality. This policy aligns with our commitment to ethical research practices and the protection of participant privacy. However, the data can be made available upon reasonable request to the IRB coordinator of icddr,b, via the following e-mail address: salamk@icddrb.org.

**Funding:** The project is jointly funded by UNICEF, Botnar Foundation, UBS Optimus Foundation, and EAGLE Foundation, Switzerland. Marufa Sultana (MS) is supported by DUPRS scholarship from Deakin University, Australia. The funders have no role in data collection, analysis, report, or writing the manuscript.

**Competing interests:** The authors have declared that no competing interests exist.

## Results

1,745 children were enrolled, 63% were male, and 57% were less than a year old. The mean societal cost per patient was US$94.2 (95% CI: US$92.2, US$96.3) with a mean length-of-stay (LoS) of 4.1 days (SD±3.0). Costs of medical personnel (US$32.6), caregiver's productivity loss (US$26) and medicines (US$22) were the major cost contributors. Mean cost was significantly higher for urban-located facilities compared to rural (difference US$17, 95% CI: US$12.6, US$20.8). No cost variation was found by age, sex, and clinical variables.

## Conclusion

Findings suggest that this novel DCA management approach is a low-cost management option, and particularly beneficial for rural residences and therefore has the potential to reduce the overall cost burden for childhood severe pneumonia management. These findings have implications for policy-making decisions in resource-poor settings for childhood pneumonia management.

## Introduction

Pneumonia remains a major cause of death among under-five children, with an estimated 0.74 million deaths globally each year [1]. While childhood mortality due to pneumonia decreased by 37% between 2005 and 2015, low and middle-income countries (LMICs) still account for 95% of pneumonia-related deaths [2,3]. Severe pneumonia requires hospitalisation with antimicrobial and other supportive treatments, and around 12% of pneumonia cases progress to this stage [3]. Evidence revealed that severe pneumonia inpatient care costs an average of US$781 per patient per episode (Range: US$39 to US$6,943 [2019 value]) in LMICs, with an average hospital length of stay (LoS) of 5.8 days, and costs increased with longer LoS [4]. However, the majority of pneumonia-related deaths occur outside of hospital settings (81%) [5], indicating that care was not sought by most of the affected families due to barriers such as accessibility and financial barriers [6–9].

Bangladesh, a LMIC with a high burden of childhood pneumonia, with an estimated 1.87 million cases each year with 21,000 thousand deaths of under-five children [10]. Only 42% of families seek care for pneumonia, but the disease accounts for 39% of total admissions, with an average hospital stay of 4 days [9,11,12]. Insufficient availability of public inpatient care facilities in Bangladesh has resulted in a significant number (22%) of children with pneumonia being denied admission [13]. Alternative management approaches are needed to provide proper care for children with severe pneumonia without hospitalization to increase families' access to care and reduce under-five mortality.

Day-care treatment approaches, that do not require 24-hour hospital care, have been successfully implemented for various conditions that typically require inpatient care [14–18]. In this approach, patients receive in-hospital treatment during daytime

hours, interspersed with periods at home with prescribed treatment and instructions. Evaluation results demonstrate comparable clinical outcomes at reduced overall costs, with the day-care approach acceptable, convenient, and more affordable for families. In Bangladesh, an innovative day-care management approach (DCA) was piloted for managing severe pneumonia in children, aiming to improve treatment effectiveness and accessibility [19,20]. This approach offers antibiotics and supportive treatment in a hospital/clinic setting during daytime hours, with continued parent-led care at home repeated each day until the clinical discharge criteria are met [19,20]. Two randomized trials compared severe pneumonia DCA management to hospital care [21,22], both concluding that when timely referral mechanisms are in place, severe pneumonia can be successfully managed in DCA (~95% cases) with lower costs compared to hospital care. However, these small-scale pilot studies were limited to the capital city of Bangladesh (an urban setting) with limited sample sizes, focusing primarily on the clinical effectiveness of the new DCA approach. As a result, gaps remain in understanding the relative costs and cost-effectiveness of this innovative intervention within existing public healthcare settings in Bangladesh covering both urban and rural contexts. To test the DCA model in a broader range, including urban and rural primary-level facilities, a cluster randomized effectiveness trial was conducted by the International Centre for Diarrhoeal Disease Research, Bangladesh (icddr,b) to assess the clinical effectiveness of DCA compared to usual care (UC) in the real-life setting. (ClinicalTrials.gov Identifier: NCT02669654) [23]. Cost analysis of trial data from the usual care arm (16 control clusters, n = 1,472), revealed that the societal cost of UC management per patient was US$195, which included household costs (US$147) [24] and healthcare provider costs (US$48) [25]. To understand the detailed cost of the intervention arm of the trial (DCA approach), it is inevitable to comprehensively measure the per-patient cost of DCA arm, enabling a comparison of both treatment approaches to determine the low-cost treatment option. As such, in the present study, a societal perspective was adopted to estimate per patient cost of DCA, as well as to examine cost variation by facility locations (urban-rural), socio-demographic and clinical characteristics, to determine cost predictors. The findings of the study will inform clinical trialists and policymakers in determining whether the new approach is a low-cost solution that improves access to care, particularly for rural residents, and has a broader impact on improving care-seeking.

## Methods

### Study design, setting, and population

This study analysed the cost of the day-care management approach for severe pneumonia in children alongside a cluster-randomized effectiveness trial conducted by icddr,b in Bangladesh. The trial compared the clinical effectiveness of DCA (intervention arm) over UC (control arm) for treating childhood severe pneumonia. Details of the clinical trial with outcomes have been published elsewhere [23]. In brief, the trial was conducted in four primary study sites, including two urban field sites in the Dhaka metropolitan area and two rural field sites. A total of 32 facilities were included in the trial, with eight clusters allocated to each study site. Children aged 2–59 months with WHO-defined severe pneumonia were enrolled between November 2015 and March 2019. For this study, the analysis focused on children enrolled in the DCA intervention clusters (n = 16 clusters across all 4 study sites).

### The DCA intervention

The details of the intervention can be found in the clinical outcomes paper [23]. Briefly, children in the DCA intervention arm received supportive treatment according to the trial protocol, which included daily intramuscular injections of ceftriaxone (75–100 mg/kg) for five days, analgesics or antipyretics as needed, a therapeutic milk-based diet, oxygen saturation measurement, nebulization, and oxygen supply, if necessary, but only during daytime hours. Treatment was provided between 8:30 am and 4:00 pm at urban facilities and between 8:30 am and 3:00 pm at rural facilities. Treatment continued every day of the week until the clinical endpoints for discharge were met, defined as no fever, no fast breathing, no tachycardia, and no hypoxemia. Children with no improvement in DCA management were referred to primary or secondary-level inpatient care facilities. Children with hypoxemia requiring prolonged oxygen were referred to the nearest primary or secondary-level inpatient hospital.

## Perspective

The perspective is crucial in conducting an economic evaluation of a health intervention, as it defines the scope of the study and determines which costs and outcomes are included in the analysis [26]. There are various types of perspectives including the patient, healthcare provider, health system, and societal [26]. The societal perspective includes all healthcare-related costs, regardless of the payer, such as patients' costs (e.g., travel, accommodation). It may also encompass non-health-related impacts in sectors such as productivity loss of the caregiver [27]. This study adopted a societal perspective to capture all costs borne by households (including caregiver productivity loss) and providers for managing one episode of childhood severe pneumonia in DCA.

## Cost measurement

A bottom-up micro-costing approach was used to identify, measure and value all relevant economic data associated with severe pneumonia treatment. This involved identifying all resources used for treatment, determining their quantity and unit price, and valuing them accordingly [26]. In estimating household costs, both direct medical costs, direct non-medical costs, and indirect costs were considered. Direct medical costs included out-of-pocket (OOP) expenses for resources such as consultations and medicines, while direct non-medical costs covered non-healthcare items like transport used for treatment purposes. Indirect costs were estimated using the human capital approach, multiplying lost working hours by the self-reported hourly wage rate, as per previous studies in similar settings, including Bangladesh [4,24,33]. For caregivers not engaged in paid work, the minimum government-fixed hourly wage rate was applied [24]. Data to estimate caregivers' indirect costs, including waiting time and lost productivity or opportunity costs, were collected through a structured questionnaire.

To estimate provider costs, we developed a comprehensive list of all resources used in one episode of care, including capital and recurrent/variable cost items such as personnel, healthcare consumables, and non-healthcare consumables. Capital item is defined as the items that is usually invested in bulk and used over time; that is, for more than one year, and recurrent item is defined as the items that are purchased regularly within a year [28]. We then measured individual-use and shared-use items. To determine the unit cost for each resource, we used the price of the item or estimated opportunity cost. For instance, capital resource items are used over multiple years, and therefore, their costs were spread over time. The equivalent annual cost (EAC) for each capital item was calculated to determine the annual cost for each asset used by the provider. EAC was derived based on the quantity, unit price, and assumed useful lifetime of each asset (5–10 years for equipment and 30 years for buildings), as indicated in the literature and warranty periods [4,24]. Data collection tools minimized double counting by collecting patient-specific data from household and provider perspectives, ensuring accurate expense allocation and cross-checking for potential duplicates. Further details of methods of measuring and valuing household and provider cost items can be found in our previous publications [24,25]. Patient-level costs were estimated by multiplying the unit cost of each item by the quantity used for each patient. These patient-level costs were then summed up to obtain the total cost for each patient. To estimate the societal cost of DCA, household costs and healthcare provider costs were combined for each child. Staff training in this study was integrated into the research-related costs associated with the trial setting and, therefore, was not included in the patient-level cost analysis. The analysis considered resource use from the period of trial enrolment date to the three–month post-discharge follow-up period, therefore discounting was not applied.

## Data collection

Trained and experienced interviewers were recruited to collect cost data from households and healthcare providers. Household and provider data collection tools were piloted in both urban and rural areas to identify any necessary modifications and potential barriers prior to final data collection.

For household data collection, resource use data, including expenses, were collected through face-to-face interviews with caregivers on the days of enrolment, discharge, and follow-up visits [24]. Provider data were collected by in-person interviews with relevant personnel at each included facility, including management staff, finance staff, administration staff,

and clinical staff. Facility-level data, such as capital items, was collected for shared cost estimation [25]. Written informed consent was obtained from the parents of all participants, as well as from the respective healthcare providers at each facility, prior to the commencement of data collection.

## Data analysis

Descriptive statistics were used to analyse resource use for each item, and baseline characteristics were presented as frequencies, proportions, and averages with 95% confidence intervals (CIs). For incomplete household cost data (~2%), mean imputation was used. This involved calculating the mean of the available data points for each variable and replacing the missing values with this mean value [29]. When cost data for healthcare facilities were unavailable, such as missing price information for equipment, the purchased price was obtained and replaced using information gathered from similar facilities for the same equipment. All analyses used a statistical significance level of $p < 0.05$ with 95% CIs. Costs were presented in United States Dollar (USD) using 2019 prices (1 USD = 84.5 BDT) and were inflation-adjusted if they were incurred prior to 2019. Various tests, including independent two-sample t-tests, $\chi2$ tests, Wilcoxon signed-rank tests, and one-way ANOVA, were used to compare cost variation across different groups and categories depending on the type of data (e.g., continuous, categorical, skewed). Cost data were skewed; therefore, the multivariable logged-linear model was used for regression analysis to determine predictors of the societal cost of DCA, with societal cost as the dependent variable and different demographic (e.g., rural/urban location, age, sex, parent education), socio-economic (e.g., wealth quintile), and pneumonia-specific clinical variables (e.g., nutritional status) as explanatory variables. The model had a mean-variance inflating factor of less than 10 meaning that there was no notable multicollinearity in this model. A one-way sensitivity analysis was performed, incorporating all pertinent cost variables, to assess the robustness of the findings. The analysis involved testing the impact of varying each cost parameter by ± 20% for both direct and indirect costs, as well as ± 1 day in average LoS at the hospital.

## Ethical Approval

The clinical trial protocol (NCT02669654) was approved by the Research Review and Ethical Review Committees of icddr,b (PR-14066). The economic evaluation protocol was approved by Deakin University Human Research Ethics Committee (DUHREC) (2018–149).

## Results

### Demographic and socio-economic characteristics of the enrolled children

The study enrolled 1,745 children in the DCA arm (Table 1). Children were mainly male (63%), with a mean age of 13.5 months (SD ± 10.8) and 57% were enrolled in rural areas. The age and sex distribution were similar across urban and rural areas, but the wealth distribution was skewed, with greater wealth observed in urban households (Table 1).

### Resource utilisation pattern

Table 2 presents the pattern of resource utilisation for treating children who received DCA treatment. The most commonly utilised healthcare resources were health personnel (e.g., physician, nurses; used by 100% of children), medicines (e.g., antibiotics; 100%), medical consumables (e.g., syringes; 100%) and nebuliser (100%). Transportation was the most commonly used non-healthcare resource (86%) and the indirect cost of caregiver time was used by 100% of children. The pattern of resource use was significantly different between urban- and rural-located DCA facilities: use of nebulisers, oxygen and transportation was higher in urban-located facilities; use of diagnostic tests was higher in rural DCA facilities (Table 2).

### Overall societal cost and its urban-rural cost variation

The estimated mean societal cost per patient per episode of childhood severe pneumonia treated with DCA model was US$94.2 (95% CI: 92.2, 96.3) [median (Interquartile Range (IQR)): 84.5 (33.5)], with a mean length of stay of 4.1 days

**Table 1. Background characteristics of enrolled children in DCA (n = 1,745).**

| Variables | Overall (n = 1,745) | Urban (n = 742) | Rural (n = 1,003) | P-value |
|---|---|---|---|---|
| **Sex, n (%)** | | | | |
| Male | 1,092 (62.6) | 457 (61.6) | 635 (63.3) | 0.46 |
| **Patient age in month, n (%)** | | | | |
| ≤ 12 m | 997 (57.1) | 423 (57.1) | 574 (57.2) | 0.23 |
| 13–24 m | 472 (27.1) | 214 (28.8) | 258 (25.7) | |
| 25–36 m | 187 (10.7) | 74 (10.1) | 113 (11.3) | |
| > 36 m | 89 (5.1) | 31 (4.2) | 58 (5.8) | |
| Patient age, mean (± SD) | 13.5 (10.8) | 13.1 (10.2) | 13.9 (11.3) | 0.12 |
| **Length of Stay (LoS) in days** | | | | |
| LoS (mean ± SD) | 4.1 (3.0) | 4.2 (3.1) | 4.0 (3.0) | |
| **Mother's education, n (%)** | | | | |
| No formal education | 260 (14.9) | 148 (20.0) | 112 (11.2) | < 0.001 |
| Up to primary | 585 (33.5) | 235 (31.7) | 350 (34.9) | |
| Secondary | 773 (44.3) | 299 (40.3) | 474 (47.3) | |
| Higher | 127 (7.3) | 60 (8.1) | 67 (6.7) | |
| **Father's education, n (%)** | | | | |
| No formal education | 418 (24.0) | 187 (25.2) | 231 (23.0) | 0.70 |
| Up to primary | 488 (28.0) | 200 (27.0) | 288 (28.7) | |
| Secondary | 677 (38.8) | 288 (38.8) | 389 (38.8) | |
| Higher | 162 (9.3) | 67 (9.1) | 95 (9.4) | |
| **Mother's occupation, n (%)** | | | | |
| Housewife | 1,475 (84.5) | 599 (88.7) | 876 (87.3) | < 0.001 |
| Informal worker (e.g., maid) | 100 (5.7) | 81 (10.9) | 19 (1.2) | |
| Service | 118 (6.8) | 31 (4.1) | 87 (8.7) | |
| Student | 21 (1.2) | 11 (1.5) | 10 (1.0) | |
| Other | 31 (1.8) | 20 (2.7) | 11 (1.0) | |
| **Father's occupation, n (%)** | | | | |
| Farmer | 183 (10.5) | 2 (0.3) | 181 (18.1) | < 0.001 |
| Informal worker | 213 (12.2) | 115 (15.5) | 98 (9.8) | |
| Transport worker | 315 (18.1) | 169 (22.8) | 146 (14.6) | |
| Salaried employee | 518 (29.7) | 234 (31.5) | 284 (28.3) | |
| Business | 355 (20.3) | 173 (23.3) | 182 (18.2) | |
| Other | 161 (9.2) | 49 (6.6) | 112 (11.2) | |
| **Household size, income and expenditures (US$, 2019), (mean ± SD)** | | | | |
| Household size | 5.4 (2.1) | 5.0 (1.8) | 5.8 (2.2) | < 0.001 |
| Average monthly income | 250.9 (269.1) | 258.7 (283.6) | 245.4 (257.7) | 0.30 |
| Average monthly expenditure | 181.0 (133.2) | 210.2 (146.2) | 159.6 (118.2) | < 0.001 |
| Average health expenditure (3 months) | 53.5 (322.5) | 53.8 (358.2) | 53.79 (293.2) | 0.96 |
| **Wealth quintile, n (%)** | | | | |
| Poorest | 410 (23.5) | 25 (2.6) | 210 (40.9) | < 0.001 |
| Poorer | 414 (23.7) | 70 (7.3) | 158 (30.8) | |
| Middle | 382 (21.9) | 151 (15.9) | 110 (21.3) | |
| Richer | 320 (18.3) | 293 (30.7) | 31 (6.0) | |
| Richest | 219 (12.6) | 415 (43.5) | 5 (1.0) | |

**Table 2. Pattern of resource use for the management of childhood severe pneumonia.**

| Parameters | Resource used n (%) | | | |
|---|---|---|---|---|
| | Overall (n = 1,745) | Urban (n = 742) | Rural (n = 1,003) | P-value |
| Medicine | 1,745 (100) | 742 (100) | 1,003 (100) | 0.983 |
| Personnel (e.g., physician, nurse) | 1,745 (100) | 742 (100) | 1,003 (100) | 0.984 |
| Diagnostic tests | 174 (10.0) | 63 (8.5) | 111 (11.1) | **<0.05** |
| Medical consumables | 1,745 (100) | 742 (100) | 1,003 (100) | 0.935 |
| Nebulization | 1,571 (90) | 729 (98.4) | 803 (80.1) | **< 0.001** |
| Oxygen | 340 (19.5) | 203 (27.6) | 137 (13.7) | **< 0.001** |
| Transportation | 1,507 (86.4) | 723 (97.4) | 784 (78.2) | **< 0.001** |
| Dietary/food | 12 (1) | 11 (1.5) | 1 (0.1) | **< 0.01** |
| Lodging/other expenses | 68 (4) | 5 (0.7) | 63 (6.3) | **< 0.001** |
| Caregiver time/productivity loss | 1,745 (100) | 742 (100) | 1,003 (100) | – |
| Capital items (e.g., furniture) | 1,745 (100) | 742 (100) | 1,003 (100) | – |
| Other variable items | 1,745 (100) | 742 (100) | 1,003 (100) | – |

(Table 3). The largest contributor to cost was personnel cost (mean: US$ 32.6, median US$24.8), accounting for 35% of the total cost, followed by caregiver's time cost (27%) [mean: US$ 25.8; median: US$ 20.4) and cost for medicines (23%). The mean healthcare provider cost per patient was US$63.3 (95% CI: 62.4, 64.4), [median (IQR): 57.3 (22.8)] while the mean household cost per patient was US$30.9 (95% CI: 29.3, 32.5) [median (IQR): 24.2 (14.2)].

The average LoS was found to be similar between urban and rural DCA facilities, with a mean of 4.2 days and 4.0 days, respectively (Table 4). The overall mean societal cost per patient differed significantly between urban and rural DCA facilities (difference US$16.7 [95% CI: US$15.9 to US$19.3], p<0.001), and this difference was observed across most of the cost components (Table 4). Personnel-related costs were significantly higher in urban DCA facilities compared to rural facilities (US$43.3 [95% CI: 41.6, 45.0]) versus (US$24.7 [95% CI: 23.9, 25.4], p<0.001). The cost for nebulisation was also significantly higher in urban DCA facilities compared to rural facilities (difference US$6.9 [95% CI: 6.7, 7.4]). No significant difference was found in caregivers' time cost for receiving treatment between urban and rural DCA facilities.

## Societal cost variation by demographic, socio-economic, and clinical characteristics

Cost variations across various socio-economic and clinical variables are presented in table 5 and Fig 1. The mean cost varied across age and socio-economic groups, with a higher cost for children aged less than one (n = 996, US$95.6) and children who belonged to the richest wealth quintile (n = 219, US$103.9). There was no significant cost variation across gender. Overall, the cost of treating children with hypoxemia was relatively higher (n = 171, US$94.6) than non-hypoxemic children (n = 1,573, US$90.3), however, the difference was not statistically significant (Table 5).

Significant cost differences were observed among LoS categories within and between groups (Fig 1). The mean societal cost per patient was higher for longer LoS in both urban and rural located health facilities (P<0.001). Cost variation by LoS categories was higher for urban-located facilities (p<0.001).

## Predictors of societal costs and cost sensitive parameters

Table 6 presents the adjusted multivariable logged linear regression model that predicts the societal cost of DCA for children with severe pneumonia. Significant factors that determined societal cost were LoS, child's age, facility location, wealth quintiles, and father's occupation. An increase of one additional treatment day in was associated with a 7% increase in log-transformed total cost (p<0.001). Belonging to the 'Richer' wealth quintile was associated with a statistically significant 5% increase in costs compared to the 'Poorest'. Considering facility location, residing in rural areas was

**Table 3. Estimated mean costs per patient per episode for management of childhood severe pneumonia (n = 1,745).**

| Cost parameter(s) | Provider cost | Household cost | Societal cost |
|---|---|---|---|
| | Mean (95% CIs) ($) | Mean (95% CIs) ($) | Mean (95% CIs) ($) |
| Medicine | 19.8 (19.3, 20.3) | 1.8 (1.6,2.1) | 21.7 (21.2, 22.2) |
| Personnel (e.g., physician, nurse) | 32.11 (31.2,33.1) | 0.5 (0.4,0.5) | 32.6 (31.6, 33.5) |
| Diagnostic tests | 0.1 (0.1,0.2) | 0.3 (0.2,0.3) | 0.4 (0.3,0.4) |
| Medical consumables | 1 (0.9,1.0) | 1.8 (1.6,2.0) | 1.1 (1.0, 1.1) |
| Nebulization | 3.5 (7.1,7.7) | 0.1 (0.0,0.1) | 3.6 (3.4, 3.9) |
| Oxygen | 0.2 (0.1,0.2) | 0.1 (0.1,0.11) | 0.2 (0.1, 0.4) |
| Transportation | – | 1.8 (1.6,1.9) | 1.8 (1.6, 1.9) |
| Dietary/food | 0.5 (0.3,0.8) | 0.1 (0.1,0.2) | 0.1 (0.1,0.2) |
| Lodging/other expenses | – | 0.5 (0.4,0.6) | 0.3 (0.2, 0.4) |
| Caregiver time/productivity loss | – | 25.8 (24.8, 27.0) | 25.8 (24.6, 27.0) |
| Capital cost | 3.9 (3.8,4.0) | – | 3.9 (3.8,4.0) |
| Variable cost | 2.6 (2.5, 2.6) | – | 2.6 (2.5, 2.6) |
| **Average cost per patient** | **63.3 (62.4, 64.4)** | **30.90 (29.3,32.5)** | **94.2 (92.2, 96.3)** |

**Table 4. Cost per patient with urban-rural cost variation by each cost parameter.**

| Cost parameters | Day care approach (DCA) (n = 1,745) | | | |
|---|---|---|---|---|
| | Urban (n = 742) | Rural (n = 1,002) | Test statistics | p-value |
| | | Mean (95% CI) | Mean (95% CI) | |
| Medicine | 21.3 (20.2, 22.3) | 22.0 (21.5, 22.4) | -1.37 | 0.168 |
| Personnel (e.g., physician, nurse) | 43.3 (41.6, 45.0) | 24.7 (23.9, 25.4) | 21.29 | **<0.001** |
| Diagnostic tests | 0.5 (0.3, 0.7) | 0.3 (0.1, 0.4) | 2.62 | **<0.01** |
| Medical consumables | 0.6 (0.5, 0.7) | 1.4 (1.3, 1.5) | -13.2 | **<0.001** |
| Nebulization | 7.6 (7.3, 7.8) | 0.7 (0.6, 0.7) | 55.92 | **<0.001** |
| Oxygen | 0.2 (0.1, 0.3) | 0.3 (0.1, 0.5) | -1.11 | 0.279 |
| Transportation | 1.5 (1.2, 1.7) | 2.4 (1.8, 2.2) | -3.77 | **<0.001** |
| Dietary/food | 0.1 (0.0, 0.2) | 0.1 (0.1, 0.2) | 1.48 | 0.139 |
| Lodging/other expenses | 0.6 (0.4, 0.7) | 0.0 (0.0) | 1.37 | **<0.01** |
| Caregiver time/productivity loss | 24.6 (23.0, 26.3) | 26.7 (25.0, 28.3) | -1.63 | 0.106 |
| Capital cost | 1.4 | 5.8 | -5.63 | **<0.001** |
| Variable/recurrent cost | 2.1 | 3.0 | -1.58 | **<0.001** |
| **Average cost per patient (US$)** | **103.8 (100.5, 107.1)** | **87.1 (84.6, 89.6)** | **8.03** | <0.001 |

associated with a statistically significant 12% decrease in costs compared to urban areas (p < 0.001). The presence of malnutrition in children with severe pneumonia was not significantly associated with increased costs.

No significant association was found between cost and disease severity, such as malnutrition or hypoxemia. Sensitivity analysis revealed that LoS, personnel, caregiver's productivity loss, and medicines were the most cost-sensitive parameters (Fig 2).

## Discussion

The present study comprehensively analysed patient-specific cost data and provided insights into the societal cost and resource utilisation associated with childhood severe pneumonia management under DCA in rural and urban areas of

Table 5. Cost variation among socio-economic and clinical characteristics US$ (n = 1,745).

| Variables | Overall (n = 1,745) | | | Urban (n = 742) | | | Rural (n = 1,003) | | |
|---|---|---|---|---|---|---|---|---|---|
| | Count | Mean (95% CI) | P-value | Count | Mean (95% CI) | P-value | Count | Mean (95% CI) | P-value |
| **Length of stay (LoS)** | | | | | | | | | |
| 1 to 3 days | 258 | 76.3 (72.9, 79.791.3) | <0.001 | 133 | 76.2 (70.5, 81.6) | <0.001 | 125 | 76.4 (72.5, 80.3) | <0.001 |
| 4 to 6 days | 1384 | 90.9 (89.5, 92.2) | | 549 | 102 (100.1, 105.2) | | 835 | 83.1 (81.9, 84.4) | |
| 6 days and more | 102 | 184.8 (162.6, 207.0) | | 60 | 175 (152.3, 199.5) | | 42 | 197 (154.3, 240.6) | |
| **Patient age in month** | | | | | | | | | |
| ≤12 m | 996 | 95.6 (92.4, 98.9) | P<0.001 | 423 | 104.1 (99.3,108.8) | <0.001 | 573 | 89.4 (85.2,93.5) | <0.001 |
| 13–24 m | 472 | 93.6 (90.6, 96.5) | | 214 | 105.1 (100.0,110.2) | | 258 | 84.1 (81.3, 86.7) | |
| 25–36 m | 187 | 90.8 (86.1, 95.6) | | 74 | 101.2 (90.6, 111.7) | | 113 | 84.1 (80.7, 87.4) | |
| > 36 m | 89 | 89.1 (83.9, 94.4) | | 31 | 98.2 (85.1, 111.3) | | 58 | 84.3 (80.4, 88.2) | |
| **Sex** | | | | | | | | | |
| Male | 1092 | 94.4 (91.8, 97.0) | 0.79 | 457 | 105.1 (100.9, 109.3) | 0.934 | 635 | 86.7 (83.6, 89.9) | 0.28 |
| Female | 652 | 93.7 (90.5, 97.3) | | 285 | 101.8 (96.4,107.1) | | 367 | 87.7 (83.4, 92.1) | |
| **Household size** | | | | | | | | | |
| Less than 4 | 260 | 95.3 (84.6, 106.0) | P<0.01 | 166 | 100.1 (93.9, 107.1) | <0.01 | 94 | 86.1 (80.5, 91.6) | <0.001 |
| 4–5 | 791 | 94.1 (88.0, 100.2) | | 350 | 103.1 (98.5, 107.5) | | 441 | 87.1 (83.1, 91.1) | |
| 6–7 | 469 | 94.5 (86.5, 102.5) | | 162 | 107.5 (99.1, 115.9) | | 307 | 87.6 (83.1, 92.2) | |
| More than 7 | 224 | 92.7 (81.2, 104.3) | | 64 | 107.5 (96.3,118.6) | | 160 | 86.8 (80.6, 93.1) | |
| **Severe pneumonia with malnutrition** | | | | | | | | | |
| Severe pneumonia (SP) | 1556 | 94.6 (90.2, 99.0) | P<0.001 | 663 | 104.1 (100.6, 107.6) | 0.542 | 893 | 87.5 (84.7,90.3) | <0.001 |
| SP with moderate malnutrition | 164 | 91.1 (77.7, 104.6) | | 70 | 101.8 (90.4, 113.1) | | 94 | 83.2 (78.7, 87.6) | |
| SP with severe malnutrition | 24 | 90.4 (55.2, 125.6) | | 9 | 96.7 (70.1, 123.3) | | 15 | 86.7 (76.6, 96.7) | |
| **Hypoxaemia** | | | | | | | | | |
| Not hypoxemic | 1573 | 90.3 (86.5, 94.1) | 0.21 | 655 | 96.7 (90.8, 102.7) | <0.001 | 918 | 83.7 (79.4, 87.9) | <0.001 |
| Hypoxemic | 171 | 94.6 (92.4, 96.9) | | 87 | 104.8 (101.1,108.4) | | 84 | 87.4 (84.7, 90.1) | |
| **Wealth quintile** | | | | | | | | | |
| Poorest | 409 | 89.7 (81.8, 97.7) | P<0.001 | 62 | 79.7 (71.3, 88.1) | <0.001 | 347 | 91.5 (87.1, 96.1) | <0.001 |
| Poorer | 414 | 86.2 (78.3, 94.0) | | 96 | 99.2 (86.5, 111.6) | | 318 | 82.2 (79.4, 85.1) | |
| Middle | 382 | 93.2 (85.0, 101.4) | | 187 | 102.1 (96.7, 107.3) | | 195 | 84.7 (79.4, 90.1) | |
| Richer | 320 | 105.0 (96.0, 113.9) | | 223 | 110.1 (103.6, 116.5) | | 97 | 93.3 (78.3, 108.3) | |
| Richest | 219 | 103.9 (93.0, 114.7) | | 174 | 109.1 (103.1, 115.1) | | 45 | 84.0 (77.4, 90.5) | |

Bangladesh. The study findings demonstrate that the mean societal cost per patient per episode of childhood severe pneumonia was US$94 with 33% of this cost borne by households ($31) and 69% by providers ($63). Productivity cost of the caregivers was identified as the major contributor to household costs and personnel cost the major contributor to provider costs. Previous studies using the same trial data for UC management reported an estimated household cost per patient of US$147 [24] and a provider cost of US$48 [25], resulting in an overall treatment cost of US$195. Although this study did not undertake a formal economic evaluation comparing clinical and cost-effectiveness between the two treatment arms (DCA and usual hospital in-patient care), our findings indicate that severe pneumonia management under the daycare model is less costly and potentially leads to substantial reductions in household expenses which would improve access to healthcare by reducing the affordability barrier.

Prior evidence conducted in other countries on diseases that commonly require inpatient care has demonstrated that the care model is clinically equivalent in most cases. Additionally, managing the disease through daycare can reduce

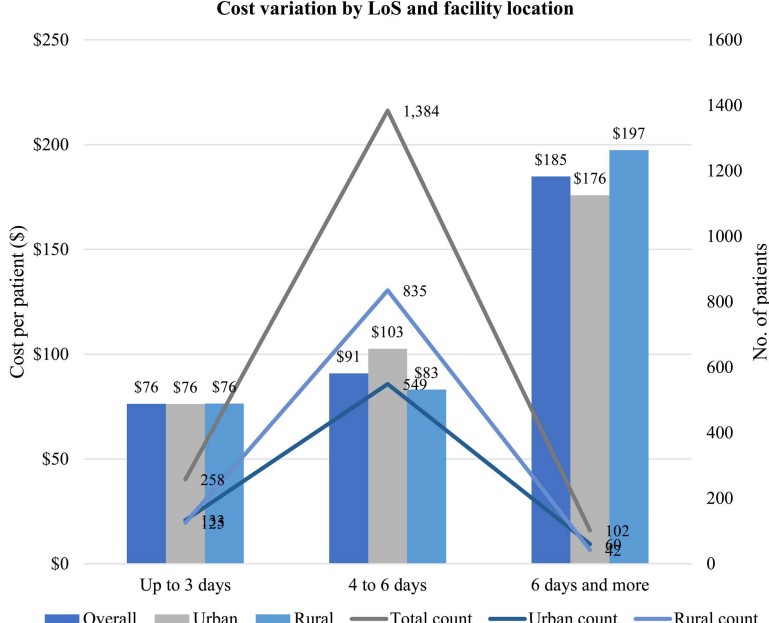

**Fig 1. Variation of mean DCA cost per patient by LoS and facility location.**

healthcare provider costs, household costs, and the burden of care for caregivers [13–18]. Previous small-scale pilot studies conducted in urban areas of Bangladesh that used the DCA model to treat severe pneumonia in children have demonstrated that the new approach is as effective as hospital care with a proper referral system and can be managed with lower costs [21,22]. Our study findings are consistent with previous research findings, which showed that patients who received treatment from DCA had lower costs and were convenient for caregivers. For instance, our study estimated the average cost of managing severe pneumonia through DCA at US$94, compared to US$114 and US$188 in previous urban-based studies [21,22]. The overall cost in our study was relatively low compared to previous small-scaled urban-based trials, which may be due to the inclusion of both rural and urban areas, with lower average costs observed in rural areas.

In resource-limited settings like Bangladesh, receiving inpatient care poses a significant financial burden on households, particularly with high OOP expenses [30]. Non-health-related expenses such as transportation and lost productivity of caregivers also act as barriers for families to access treatment [9,31–34]. For instance, hospitalisation of children in Bangladesh requires a considerable amount of caregiving time from families, as caregivers need to stay with their child in public inpatient healthcare facilities [35–37]. The current study found that the DCA model for childhood severe pneumonia management had relatively lower costs for caregiver time (US$26) and transportation (US$1.8). The reduced opportunity cost for DCA can be explained by the shorter time required from caregivers to receive treatment in a day care setting (8 hours instead of 24 hours) for their child. Similarly, the low transportation cost in DCA group indicates that making the service available close to the residence has the potential to reduce transportation costs and therefore lower opportunity costs by minimizing travel time. Therefore, it can be expected that the availability of day-based healthcare services for severe pneumonia management in proximity to residents would have mitigated distance-related barriers and would facilitate early care seeking, eventually will be beneficial for both the health system and the society.

In our study, personnel costs were found to be the primary cost driver for the DCA model, accounting for US$33 (35%) of the total. In line with our findings, a study undertaken in Pakistan reported high personnel costs (48% of the total cost)

**Table 6. Multivariate regression model.**

| Variables | Adjusted model (Overall, N = 1,745) | |
| --- | --- | --- |
| | Coef. (95% CIs) | P-value |
| Length of stay (LoS) | 0.07 (0.05,0.08) | <0.001 |
| Patient age in month | 0.01 (0.0, 0.01) | <0.05 |
| **Wealth quintile** | | |
| Poorest (ref) | | |
| Poorer | -0.03 (-0.08, 0.01) | 0.06 |
| Middle | 0.02 (-0.03, 0.06) | 0.48 |
| Richer | 0.05 (0.01, 0.02) | <0.05 |
| Richest | 0.01 (0.02, 0.12) | <0.01 |
| **Place of residence** | | |
| Urban (ref) | | |
| Rural | -0.11 (-0.15, -0.08) | <0.001 |
| **Father occupations** | | |
| Farmer (ref) | | |
| Transport workers | -0.06 (-0.11, -0.01) | <0.01 |
| Informal workers | -0.08 (-0.13, -0.03) | <0.05 |
| Salaried employees | -0.03 (-0.08, 0.01) | 0.15 |
| Business | -0.02 (-0.06, 0.03) | 0.51 |
| Others | -0.09 (-0.15, -0.04) | <0.001 |
| **Presence of malnutrition** | | |
| Not malnourished (ref) | | |
| Moderate malnutrition | -0.02 (-0.08, 0.04) | 0.60 |
| Severe malnutrition | 0.01 (-0.11, 0.09) | 0.82 |
| **Presence of hypoxemia** | | |
| Not hypoxemic (ref) | | |
| Hypoxemic | -0.01 (-0.05, 0.03) | 0.70 |
| Constant | 4.30 | – |
| Prob>F | <0.001 | |
| R-squared | 0.38 | |
| Root MSE | 0.28 | |
| Mean VIF | 1.70 | |

for inpatient care severe pneumonia management, which is slightly higher compared to the findings from our study [37]. While comparing urban and rural cost differences, a statistically significant difference was observed in personnel costs, which were substantially higher in urban DCA facilities (US$43) compared to rural facilities (US$25). This difference can be attributed to the larger number of staff, higher salaries, and the availability of more experienced and specialised personnel in urban settings, which are associated with higher costs. Urban treatment facilities often provide more effective care due to access to advanced infrastructure and specialised expertise. In contrast, rural facilities demonstrated statistically significant higher capital costs, which can be explained by the need for additional equipment and infrastructure to support continuous and protocol-compliant treatment for severe pneumonia. Cost for medicine was similar for both urban and rural DCA facilities (US$21 and US$22) for treating severe pneumonia patients which is comparable with other estimations reported by India for a secondary-level hospital (US$7) [38] and lower than estimated in Pakistan (US$42) in similar settings [34]. The average cost of consumables was higher in rural facilities. This might be explained by the fact that there are some specific medical supplies received by rural primary health facilities as a strategic goal of Bangladesh

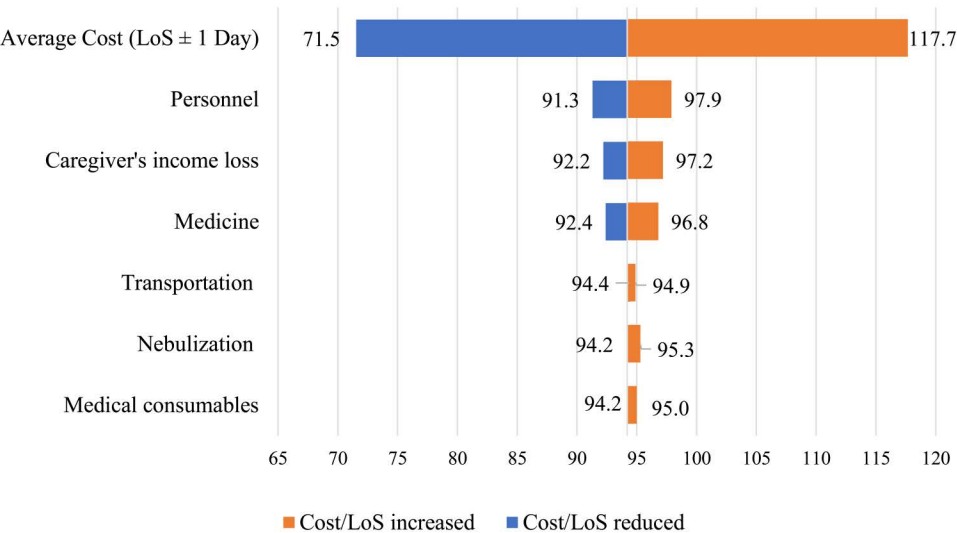

**Fig 2. Sensitivity analysis (tornado diagram).**

and hence the cost is relatively high. In addition, for DCA clusters, rural located facilities were equipped to provide standard management of severe pneumonia which might be reflected by relatively high provider costs for consumables.

## Future implications of the study

Previous published evidence highlighted the significance of financial and non-financial impediments in seeking care for under-five children in LMICs adopting both qualitative and quantitative research and suggests that accessibility and financial barriers, such as long waiting times, long distance to a healthcare facility, lack of transport, high OOP expenses, socio-economic and socio-cultural position are associated with delayed/abandoned care-seeking, specifically in rural areas [8,9,32,33,39,40].

In our study, the mean per patient cost of the DCA model shows that travel costs and caregiver time/productivity costs are lower using the model for childhood severe pneumonia management in both urban and rural located facilities, compared to the available published literature on inpatient care for severe pneumonia [21,22,24,25]. This highlights the strong argument that implementing the DCA model could significantly reduce both supply and demand-side barriers in both urban and rural areas. While the costs for medicine and supportive care to treat children with severe pneumonia may be similar across urban and rural settings, implementing this approach in rural areas would be less costly for the government, would substantially improve healthcare access, and could save children's lives by preventing them from being left untreated at home due to demand and supply-side barriers. In addition, addressing this disparity by strengthening rural healthcare services could also promote more equitable access to care while potentially reducing the reliance on costly inpatient treatments if manageable by DCA settings.

## Strength and limitation

The study has several strengths, including a large trial setting and individual-level data collection, enabling accurate measurement of resource use items. The comprehensive collection of patient-specific cost data provides a detailed understanding of resource utilization. Despite these strengths, the study has limitations. Due to constraints in data availability,

some resource use items, such as capital items, had to be estimated using a top-down approach. This may introduce some degree of uncertainty into the analysis. Additionally, the study does not consider service use from referred facilities (if not listed by trial settings). This exclusion may limit the comprehensiveness of the analysis, as it does not capture the full scope of resource utilization within the healthcare system.

## Conclusion

The study findings reveal that the societal cost of DCA for the management of childhood severe pneumonia is relatively low-cost treatment approach. The estimation revealed that personnel, caregiver productivity loss, and medicines were the major cost contributors for childhood severe pneumonia management. Factors such as length of stay, age of children, facility location, and wealth quintiles were identified as important cost predictors. Findings highlight the potential of this new approach to reduce overall household and provider cost burden. High enrolment in rural areas indicates that improving access to healthcare could improve care-seeking behaviour among vulnerable populations who may otherwise abandon seeking care. Detailed economic evaluation comparing clinical and economic evaluation is recommended to compare cost-effectiveness of DCA over UC and to determine cost-effective option to allocate scarce resources efficiently and to contribute toward inequity in distribution of healthcare services.

AcknowledgmentWe would like to acknowledge the dedicated efforts and support of the data collectors and field supervisors, who played a crucial role during the collection of patient-specific cost data. icddr,b is grateful to the Governments of Bangladesh, Canada, Sweden, and the United Kingdom for providing core/unrestricted support.

## Author contributions

**Conceptualization:** Marufa Sultana, Nur H Alam, Lisa Gold.

**Data curation:** Marufa Sultana, Nausad Ali, Sabiha Nasrin.

**Formal analysis:** Marufa Sultana, Nausad Ali.

**Funding acquisition:** Nur H Alam, George J Fuchs, Niklaus Gyr.

**Investigation:** Nausad Ali, Niklaus Gyr.

**Methodology:** Marufa Sultana, Jennifer Watts, Julie Abimanyi-Ochom, Lisa Gold.

**Project administration:** Nur H Alam, Nausad Ali, Abu S Faruque, Sabiha Nasrin, Mohammod J Chisti.

**Resources:** Nur H Alam, Abu S Faruque, Sabiha Nasrin.

**Supervision:** Jennifer Watts, Julie Abimanyi-Ochom, Lisa Gold.

**Validation:** Jennifer Watts, Nur H Alam, Abu S Faruque, Mohammod J Chisti, George J Fuchs, Niklaus Gyr, Tahmeed Ahmed, Lisa Gold.

**Writing – original draft:** Marufa Sultana.

**Writing – review & editing:** Jennifer Watts, Nur H Alam, Nausad Ali, Abu S Faruque, Sabiha Nasrin, Mohammod J Chisti, George J Fuchs, Niklaus Gyr, Tahmeed Ahmed, Julie Abimanyi-Ochom, Lisa Gold.

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
