## [Decision Letter · Decision Letter 0]

4 Dec 2024

PONE-D-24-44140Societal cost of day-care approach (DCA): a low-cost approach in urban and rural settings for management of childhood severe pneumonia in BangladeshPLOS ONE

Dear Dr. Sultana,

Thank you for submitting your manuscript to PLOS ONE. After careful consideration, we feel that it has merit but does not fully meet PLOS ONE’s publication criteria as it currently stands. Therefore, we invite you to submit a revised version of the manuscript that addresses the points raised during the review process.

**ACADEMIC EDITOR: comments ** 

The topic says “societal costs of DCA”…

What does societal costs mean?Even though it says societal costs, what is presented in the findings is economic costs. What makes it different from economic costs, which have already been studied and confirmed feasible?The feasibility for DCA is already confirmed to be feasible in Bangladesh. You mentioned this in previous RC studies. So, what was the importance of your study? What value will it add or which gap will it fill?One of the findings indicates Rural setting is more feasible for DCA than an urban setting, right? How could you justify this finding in scientific ways? Is that because of certain biases (e.g. urban people pay more than rural people do or ????), better if you can justify them scientifically?The way the indirect costs and opportunistic costs were valued/given price is not clear.In the statistical analysis part, p values only indicate significance level. But, the direction of the association is not clear. E.g. what indicates if cost is higher among urban compared to rural?You mentioned as the you collected both qualitative and quantitative data. So, what type of qualitative data, to answer what and  where are the findings from qualitative data

We look forward to receiving your revised manuscript.

Kind regards,

Seifadin Ahmed Shallo, MPH

Academic Editor

PLOS ONE

Journal Requirements:

4. In the online submission form, you indicated that the data utilised in this study cannot be shared publicly to protect the privacy of the participants. Data can be made available upon reasonable request to the corresponding author. 

5. Please review your reference list to ensure that it is complete and correct. If you have cited papers that have been retracted, please include the rationale for doing so in the manuscript text, or remove these references and replace them with relevant current references. Any changes to the reference list should be mentioned in the rebuttal letter that accompanies your revised manuscript. If you need to cite a retracted article, indicate the article’s retracted status in the References list and also include a citation and full reference for the retraction notice

Reviewers' comments:

Reviewer's Responses to Questions

**Comments to the Author**

1. Is the manuscript technically sound, and do the data support the conclusions?

Reviewer #1: Yes

2. Has the statistical analysis been performed appropriately and rigorously? 

Reviewer #1: Yes

3. Have the authors made all data underlying the findings in their manuscript fully available?

Reviewer #1: Yes

4. Is the manuscript presented in an intelligible fashion and written in standard English?

Reviewer #1: Yes

5. Review Comments to the Author

Reviewer #1: Dear authors,

Thank you. This manuscript undertakes a straightforward analysis of ‘Societal cost of day-care approach (DCA): a low-cost approach in urban and rural settings for management of childhood severe pneumonia in Bangladesh’. There are a few little fixes that would, in my view, improve the manuscript for "consumption" by other researchers. This is the new, innovative and important research in developing country context.

Comments:

Comment- 1: In an economic sense: median cost was more effective than mean. If you include both the mean and median cost, I think you should be included in the median and mean cost.

Comment- 2: Why did you suggest DCA intervention is low cost? What do you mean by resource- poor setting in pneumonia management? Are urban and rural facilities truly comparable? Otherwise, urban and rural costs might be significantly different?

Comment- 3: Please see the line- 104 to 106. Your estimation might be focused on cost effectiveness analysis. But not focused on low-cost interventions.

Comment- 4: Please see the line- 111 to 114. Please describe the true comparable facilities of DCA and UC?

Comment- 5: Please see the line- 136 to 143. All direct costs should not be OOPE. When do you estimate the opportunity cost in provider perspectives? Please describe annuitization of capital items and how you should estimation using inflation adjustment.

Comment- 6: In provider cost, you should be change in variable cost. I think it is the recurrent cost. In provider cost, I have not seen the training and meeting related cost? How do you consider opportunity cost in provider perspective?

6. PLOS authors have the option to publish the peer review history of their article (what does this mean? ). If published, this will include your full peer review and any attached files.

**Do you want your identity to be public for this peer review?** For information about this choice, including consent withdrawal, please see our Privacy Policy .

Reviewer #1: No

---

## [Editor Report · Decision Letter 1]

8 Apr 2025

Societal cost of day-care approach (DCA): a low-cost approach in urban and rural settings for management of childhood severe pneumonia in Bangladesh

PONE-D-24-44140R1

Dear Dr. Sultana,

We’re pleased to inform you that your manuscript has been judged scientifically suitable for publication and will be formally accepted for publication once it meets all outstanding technical requirements.

Kind regards,

Seifadin Ahmed Shallo, MPH

Academic Editor

PLOS ONE
---

## [Editor Report · Acceptance letter]

PONE-D-24-44140R1

PLOS ONE

Dear Dr. Sultana,

I'm pleased to inform you that your manuscript has been deemed suitable for publication in PLOS ONE. Congratulations! Your manuscript is now being handed over to our production team.

Kind regards,

on behalf of

Prof. Seifadin Ahmed Shallo

Academic Editor

PLOS ONE